# Impact of Heat Stress on Bovine Sperm Quality and Competence

**DOI:** 10.3390/ani12080975

**Published:** 2022-04-09

**Authors:** Luís Capela, Inês Leites, Ricardo Romão, Luís Lopes-da-Costa, Rosa M. Lino Neto Pereira

**Affiliations:** 1Biotechnology and Genetic Resources Unit, National Institute of Agrarian and Veterinarian Research (INIAV), 2005-048 Santarém, Portugal; lcapela@fmv.ulisboa.pt (L.C.); rosalnp@gmail.com (R.M.L.N.P.); 2CIISA, Faculty of Veterinary Medicine, University of Lisbon, 1300-477 Lisboa, Portugal; inescostaleites@gmail.com; 3Associate Laboratory for Animal and Veterinary Sciences (AL4AnimalS), 1300-477 Lisboa, Portugal; 4MED, University of Évora, 7006-554 Évora, Portugal; rjromao@uevora.pt

**Keywords:** heat stress, bovine, spermatozoa, spermatogenesis, fertility

## Abstract

**Simple Summary:**

Livestock farming is facing growing challenges due to climate change. Both animal welfare and male fertility are parameters easily affected by thermal stress. Spermatogenesis is a highly thermosensitive process, and therefore heat stress leads to a marked decrease in the fertilizing capacity of spermatozoa. There is an urgent need to find sustainable ways of mitigating the effects of global warming on animal welfare and fertility, and consequently protect the profitability of farms.

**Abstract:**

Global warming has negatively influenced animal production performance, in addition to animal well-being and welfare, consequently impairing the economic sustainability of the livestock industry. Heat stress impact on male fertility is complex and multifactorial, with the fertilizing ability of spermatozoa affected by several pathways. Among the most significative changes are the increase in and accumulation of reactive oxygen species (ROS) causing lipid peroxidation and motility impairment. The exposure of DNA during the cell division of spermatogenesis makes it vulnerable to both ROS and apoptotic enzymes, while the subsequent post-meiotic DNA condensation makes restoration impossible, harming later embryonic development. Mitochondria are also susceptible to the loss of membrane potential and electron leakage during oxidative phosphorylation, lowering their energy production capacity under heat stress. Although cells are equipped with defense mechanisms against heat stress, heat insults that are too intense lead to cell death. Heat shock proteins (HSP) belong to a thermostable and stress-induced protein family, which eliminate protein clusters and are essential to proteostasis under heat stress. This review focuses on effects of heat stress on sperm quality and on the mechanisms leading to defective sperm under heat stress.

## 1. Introduction—The Climate Change Scenario

It is now scientifically acknowledged that our planet is facing global climate change, namely global warming [1], with the Iberian peninsula, already seasonally challenged by heat stress, being one of the most affected regions [2]. Global warming is of utmost concern for livestock production and animal welfare and wellbeing; in affected regions, heat stress is the main factor responsible for decreases in fertility and productivity [3]. Since the industrial revolution, the impact of human action has led to a rise in global temperature [1]. From 1950 onwards, this rise exceeded 1 °C, with the last 7 years ranking as the hottest years since the availability of records [4]. Besides a decline on biodiversity, the loss of ecosystems and the degradation of soils, global warming has led to a significant decrease in freshwater availability [5]. Intensive and extensive cattle production systems struggle with the rising intensity and duration of heatwaves and draughts, particularly in the southern regions of Africa and the Mediterranean basin [2]. This can lead to the loss of biodiversity, as some breeds are threatened by a lack of adaptive capacity to climate change. Cattle genetic selection has mainly been based on production traits [6]. However, this has led to a loss of overall rusticity and resistance to disease, and increased sensitivity to environmental temperatures [6,7,8].

Due to this growing challenge, during the last decades, the relationships between heat stress, behavior, coat and skin particularities, thermoregulatory ability, and fertility have received increased attention [9]. Male fertility is a complex condition dependent on a broad range of environmental and genetic factors [10]. The current harshening of heat stress scenarios may become a main factor affecting bull fertility, namely in extensive beef production systems. The pathway from spermatogonia to spermatozoa includes many cellular divisions, such as the loss of cellular machinery and high DNA condensation, making this a process highly sensitive to external threats, namely temperature [11,12,13]. However, the relationship between temperature and spermatogenesis is controversial, and unanswered questions still remain. This review focuses on the effects of heat stress on sperm quality and on the mechanisms leading to defective sperm under heat stress.

## 2. Adaptive Mechanisms to Heat Stress in Cattle

According to Bligh [14], the thermal comfort zone is defined as the range of environmental temperatures in which basal metabolism is at its lowest and thermoregulation is achieved without evapotranspiration. Most livestock species set their thermal comfort zone between 16 and 25 °C [15]. In cattle, perspiration is the first activated thermoregulatory mechanism, initiated in dairy cows at around 14 °C, well before the most visible sign, tachypnea [16], and subsequent increased water intake [17] and decreased food ingestion and productivity [18]. Moderate-to-severe heat stress leads to increased metabolic requirements, ranging from 7% to 25%, depending on the amount of energy used to cool down the animal [19]. Thyroid T3 and T4 blood concentrations are reduced to decrease basal metabolism [20], a remarkable adaptive mechanism best observed in individuals better adapted to heat stress [21]. Additionally, adipocytes increase the synthesis of leptin and adiponectin, which reduce ingestion [22].

Over 24 °C, the internal body temperature rises to levels that render the animal’s thermoregulatory mechanisms insufficient [23,24]. The impact of humidity on the thermoregulatory capacity of animals relates to their capacity to sweat. Cattle tolerate high temperatures in the absence of high relative humidity because they are able to efficiently release heat via evapotranspiration [25]. The most commonly used indicator for thermal stress is the temperature–humidity index (THI), which takes into account air temperature and relative humidity, through several developed formulas [26,27,28]. *Bos taurus* enter thermal stress at THI 70, with thermoregulatory mechanisms ceasing to be effective above THI 80 [29,30]. Zebus are known for their resistance to heat, initiating heat-related tachypnea and perspiration at temperatures 8 °C higher than *Bos taurus* [31]. Blood and saliva cortisol concentrations have been used to assess stress caused by transport [32], castration [33], and dehorning [34]. However, sampling requires handling of the animal, which is itself stressful, and results represent concentrations at the time of sampling, potentially including the risk of an induced increase in cortisol levels [35]. Hair cortisol levels, in contrast, can assess chronic or long-term stress [36], such as thermal stress, related to the previous several months [37], as cortisol is incorporated into hair growth in a continuous way [38].

Heat can disrupt testicular thermoregulation, interrupting and impairing spermatogenesis and thus leading to transitory subfertile or infertile animals with poor semen quality [39]. Mammalian spermatogenesis occurs at 4 °C to 5 °C below body temperature and depends of the thermoregulatory ability of the testis, which are located outside body cavities, an evolutionary step that emphasizes the importance of temperature for physiologic spermatogenesis [40,41]. *Bos indicus* breeds are known for their resistance to hot weather and have developed adaptive mechanisms of testicular thermoregulation, including a smaller testicular artery wall thickness and arterial–venous blood distance. *Bos taurus* have also developed adaptation mechanisms to restore homeostasis under heat stress, namely, feed intake and T3/T4-based metabolism adjustments, increased heat loss ability, less intense tachypnea, and a more efficient sweating rate [2,3]. Heat-resistant bulls show a more pendulous scrotum and increased testicular length compared to heat-sensitive bulls [42]. Additionally, heat-resistant bulls display a scrotal and testicular subtunic positive temperature gradient, whereas in heat-sensitive bulls this is only observed in the scrotum. The above characteristics of the testicular vascular cone may largely account for the resistance to heat in bulls [42]. Response to heat stress depends on factors such as age, body composition (fat distribution), breed (genotype), and environmental conditions (wind, shadow, and water availability) [43]. However, the extent of heat stress-induced damage is also related to individual thermotolerance ability, as shown in boars [44].

## 3. Cell Response to Heat Stress

Under heat stress, the accumulation of ROS in mitochondria leads to the oxidation of I, II, III and IV complexes, limiting oxidative phosphorylation, and therefore reducing ATP synthesis and lowering the cell basal metabolism [45,46]. The mechanism by which heat stress leads to oxidative stress remains unclear, but an imbalance between pro-oxidative and anti-oxidative compounds is noticeable, leading to high levels of anti-oxidative enzymes, such as dismutase, catalase and glutathione peroxidase and, concomitantly, low levels of anti-oxidative compounds, such as beta-carotenes and vitamins E, C, and A [15,46]. Acute hyperthermia, with temperatures around 45.5 °C, has been found to lead to acute cell death by necrosis [47], whereas at temperatures around 42 °C the heat insult induces cell death slowly or rapidly. In slow-mode cell death, there is a delayed damage of centrosomes. When mitosis starts, this leads to the so-called “mitotic catastrophe”, as damaged centrosomes cause multipolar cell division, and subsequent death by apoptosis [48]. In rapid-mode cell death, apoptosis and necroptosis lead to cell death within hours after the heat insult. In both cell death modes, receptor interaction protein kinase (RIPK1) seems to play an important role, controlling cell survival pathways such as apoptosis or necroptosis [49]. The cell cycle is also affected by exposure to heat stress, with a significant reduction in cell multiplication due to the arrest in cell phases S [50], G1/S, and G2/M [51].

### Heat Shock Proteins (HSPs)

Heat-stressed cells exhibit a robust HSP production. This protein family is classified by their molecular size, consisting of seven major groups: Hsp110, Hsp100, Hsp90, Hsp70, Hsp60, Hsp40, and the 15–30 KDa HSPs, named small HSPs [52]. HSPs are ubiquitous proteins essential to proteostasis, most being stress-inducible with multiple chaperone functions such as protein trafficking, folding and refolding, protein complex disaggregation, and degradation of misfolding proteins by the ubiquitin proteolysis pathway [53,54]. HSPs are mainly activated by oxidative stress, a mechanism essential to cell survival, protecting enzymes and organelles from oxidation [55]. Most HSPs are intracellular molecules, but some show an extracellular immunomodulatory function too. HSP60 can be found in the bloodstream, as a pro-inflammatory (high concentration) or anti-inflammatory (low concentration) signal to macrophages and dendritic cells [56]. HSP70, present in the extracellular matrix, triggers the immune system, inducing TNF-α and facilitating antigen presentation to T cells [57]. Some HSPs work together: HSP27, HSP70, and HSP90 prevent cell death during heat stress by blocking different steps of the apoptosis cascade [58]. HSPs also have a structural function in cell shape and membrane stabilization. During hyperthermia, cell and organelle membranes lose their structure and become more fluid due to the increased saturation of lipids and the formation of nonpolar hexagonal structures in the lipidic bilayer [59]. This disrupts the control of cell volume and membrane potential, and in response to this injury, cells increasingly include HSPs in the membrane, which restores normal fluidity [59,60].

HSPs play an essential role in spermatogenesis. For example, HSP90α-deficient male mice are sterile due to a complete failure to produce sperm. This protein is critical for the first wave of spermatogenesis before puberty and for the maintenance of spermatogenesis in adult testis [61,62]. A role for HSP90 in the motility of bull post-thawed sperm was reported [63], and rams presenting different genotypes in the HSP90AA1 promoter showed differential responses of sperm to heat stress [64].

## 4. Impact of Heat Stress on Spermatogenesis

There is a temperature threshold, also influenced by duration of insult, above which germ cell degeneration is induced, as demonstrated in mice [65]. Exposing male mice to elevated THI affects sperm morphology and plasma membrane integrity, kinematics and concentration, increasing sperm mortality and leading to infertility [66,67,68,69]. Heat stress increases testicular temperature, leading to sperm abnormalities (Figure 1). The proportion, severity, and moment of appearance of abnormal sperm in the ejaculate depend on the intensity and duration of heat stress, and the developmental stages of affected germ cells [70]. Predominant abnormalities are located in the sperm head (acrosome defects, pyriform-shaped heads, micro and macro-cephalic heads) and tail [66]. Mild, short-term heat stress can significantly affect sperm motility 14 to 21 days after exposure, as a result of insult to the spermatid and spermatocyte stages of development, where formation of the flagellum occurs [13].

### 4.1. Oxidative Stress

An increase in testicular temperature increases testicular metabolism and oxygen consumption to sustain aerobic metabolism [70]. When this is not matched by a sufficient blood flow to maintain an adequate level of oxygenation, testicular tissue can become hypoxic, leading to tissue oxidative stress [70,71]. However, this concept was recently challenged by Rizotto and Kastelic [72], who found no evidence of heat stress-induced hypoxia in the testis. In several animal models, heat stress-induced low sperm quality has been related to an increment in ROS production and subsequent lipid peroxidation [73] and DNA damage [65,74,75]. Sperm lipid peroxidation results in the loss of plasma membrane integrity, mitochondrial membrane dysfunction (impairing energy production and sperm motility), chromatin degradation [65,74], and sperm morphological abnormalities [76]. Seminal plasma lipids are less susceptible to peroxidation compared to sperm lipid membranes [39].

Bovine sperm motility is very sensitive to oxidative stress induced by hydrogen peroxide due to the effect of ROS on the contractility mechanisms of the sperm tail [77]. The mitochondrial regulation of redox balance is crucial for cell homeostasis. Apoptotic mechanisms activated by oxidative stress are initiated following an imbalance between ROS production and antioxidant ROS neutralization [15,73,78]. Spermatozoa are highly vulnerable to oxidative stress because, due to having limited cytoplasm to house an appropriate amount of defensive enzymes, they lack antioxidant capacity. Physiologically, sperm mitochondria are prone to electron leakage during oxidative phosphorylation (OXPHOS) leading to ROS generation [71,73], and this OXPHOS occurring in the inner mitochondrial membrane and cristae is essential for energy production. Depending on species and environmental conditions, sperm cells produce energy through OXPHOS and glycolysis. OXPHOS is predominant in bovine spermatozoa, and the majority of energy is directed towards motility [79,80]. However, under heat stress, mitochondrial capacity is compromised and ROS production rate increases [73,81]. In ruminants and humans, acute and chronic exposure to heat stress were found to increase sperm mitochondrial ROS generation, sperm membrane fluidity, and oxidative DNA damage [75], impairing in vitro fertilization and embryo production [82,83]. Mitochondrial respiration rates show variation among individuals under heat stress. Sperm cells with low mitochondria respiratory capacity were found to produce more hydrogen peroxide, and males were less tolerant to temperature changes, underlining the relevance of mitochondria in setting thermal tolerance limits [81]. Different strategies to minimize ROS deleterious effect are currently under research, including anti-oxidants directed at mitochondria [84,85,86]. Sperm oxidative status can be evaluated by flow cytometry using cellRox and 2′,7′-Dichlorofluorescein Diacetate (DCFH) [77,87].

### 4.2. Apoptosis

Heat stress induces male germ cell mitotic catastrophe and apoptosis, conveying a reduction in testicular weight and seminiferous tubule size [13,41,48]. This apoptotic process occurs by intrinsic and extrinsic molecular pathways [74,88], in which the main intracellular effectors are caspases, a family of cysteine proteases, acting as executioners (caspase 9) and initiators (caspases 3, 6 and 7) [88,89]. The apoptotic intrinsic pathway involves the activation of Bcl-2 family members, including a redistribution of Bax from its cytoplasm to a paranuclear location, followed by the sequestration of ultracondensed mitochondria and endoplasmic reticulum to these paranuclear sites [88,89,90]. Proteins are released from mitochondria to the cytosol, and cytochrome C and DIABLO have emerged as regulators of this pathway in several male models [88,89,90,91]. The apoptotic extrinsic pathway involves ligation of a death receptor (Fas) to its ligant (FasL), ultimately recruiting FADD (Fas-associated death domain) [92,93,94]. Both the intrinsic and extrinsic pathways converge on caspase 3 and other executioner caspases and nucleases that trigger the events of programmed cell death [88]. The evaluation of apoptosis in the testicular parenchyma and spermatozoa of several species has been widely accomplished by TUNEL assay [95,96,97].

### 4.3. DNA Damage

The testis parenchyma comprises germ cells in all stages of development, as well as somatic cells (such as Sertoli and Leydig). Under heat stress, gene transcription and transduction are impaired, which affects the germline progression and synthesis of male hormones [67,89]. In fact, heat stress-induced sperm DNA damage is not only caused by the increased generation of ROS, but also by an impairment in gene expression related to defense mechanisms against DNA damage [75,89]. Germ cells are sensitive not only to high and long-term heat stress, but also to short-term heat stress; a temperature of 43 °C for 5 min to 45 min induces DNA fragmentation compatible with an apoptotic pattern, as demonstrated in mice [11]. Within the seminiferous spermatogenic cycle, the spermatocyte and spermatid stages are the most temperature sensitive [65,67]. Spermatocytes II, as post-meiotic cells, and spermatids, following chromatin condensation, lose apoptotic response and DNA damage repair ability [66,67]. Recently [12,98], DNA damage was localized to regions with low chromatin compaction, namely where protamines are replaced by histones, which have been identified as DNA oxidation-vulnerable regions. In mice, germ cell loss is accompanied by the formation of degenerated giant germ cells and ‘gaps’ on the seminiferous epithelium and, despite apoptosis machinery, many of these damaged cells proceed into sperm with damaged DNA [65]. In this way, it is possible that damaged DNA sperm cells can achieve fertilization but later lead to defective embryo development and subsequent embryo or fetal loss [65]. Sperm DNA fragmentation can be assessed through TUNEL [97], and chromatin stability can be evaluated by a chromatin structure assay [99].

In rodent early embryos, heat stress may induce sperm germ cell DNA mutation or deletion leading to the disruption of regulatory networks [65], namely the expression of paternal imprinted genes, inhibiting subsequent embryo development [100]. In bovines, the strategic placement of histones in male germ cells, a crucial epigenetic marker of parental DNA, may be affected and lead to embryonic epigenetic modifications, which impair blastocyst development [69]. The sperm epigenome is acquired from the embryonic stage until after puberty. During spermatogenic cycles, exposure to heat stress has an impact on DNA methylation, affecting male pronucleus formation, with a subsequent decrease in embryonic developmental potential [101,102]. This continual epigenetic remodeling state makes sperm cells very susceptible to environmental factors, but differences in their epigenetic profiles may be related to different heat stress susceptibilities demanding further research. Furthermore, spermatozoa RNA involved in zygote cleavage may be targeted by oxidative stress, which can be deleterious even before embryo genome activation [77].

## 5. Heat Stress Effects on Spermatozoa Capacitation and Fertilization Ability

Capacitation, a spermatozoa maturational requirement for fertilization, comprises a state of physiological oxidative stress involving cholesterol efflux from the plasma membrane, an increase in intracellular cAMP and Ca^2+^ concentrations and in tyrosine phosphorylation [103,104]. Heat stress can impair sperm motility 2 weeks after heat exposure, decreasing progressive motility to 40%, which may only recover from 8 weeks following the insult [72]. In fact, spermatozoa are strong ROS generators, which is a major pathway for capacitation via the redox regulation of tyrosine phosphorylation [104] and actin polymerization via cAMP/PKA [105]. However, this redox-regulated capacitation cascade may damage spermatozoa due to excessive oxidative stress. When the capacitation period is prolonged, spermatozoa enter an ‘over-capacitation’ state, which culminates in the enhanced release of mitochondrial ROS and cell death [106].

Heat stress-induced oxidative stress promotes premature capacitation and an increase in the proportion of capacitated and acrosome-reacted spermatozoa [100]. This state of premature capacitation reduces sperm lifespan and fertilization and embryo development rates. Additionally, sperm cells may carry metabolites that induce oocyte damage, such as lipid peroxidation and antioxidant depletion, which may impair embryo development [77]. The capacitation status of sperm is historically assessed by the chlortetracycline assay, whereas fertilization and embryo development rates can be evaluated through in vitro assays [77].

## 6. Conclusions

It is known that heat stress impairs spermatogenesis, decreases sperm motility and concentration, and increases the number of defective sperm, leading to low fertilization and embryo development rates, ultimately resulting in subfertility or infertility. Heat stress heavily compromises bull breeding performance. Environmental cooling has been explored, but there has been no practical application in extensive beef production systems. One way to mitigate the effects of heat stress in cattle fertility may be accomplished by the genetic selection of more adapted and thermo-tolerant animals. Another way to face heat stress effects is the deciphering of cellular and molecular mechanisms underneath the deleterious effects and the development of appropriate control measures. Future approaches may well rely on a combination of these two approaches. The feed incorporation of antioxidant compounds aiming to reduce ROS accumulation and the formulation of less thermogenic diets are under study [107,108]. However, the number and quality of HSPs and cellular DNA repair capacity mechanisms remain individual factors, making the genetic selection of thermotolerant animals a feasible goal to control heat stress-induced male infertility. In the present scenario of climate change and increasing societal pressure for efficient and environmentally friendly livestock farming, the scientific community urgently need to mitigate heat stress effects on animal welfare, fertility, and production.

## Figures and Tables

**Figure 1 animals-12-00975-f001:**
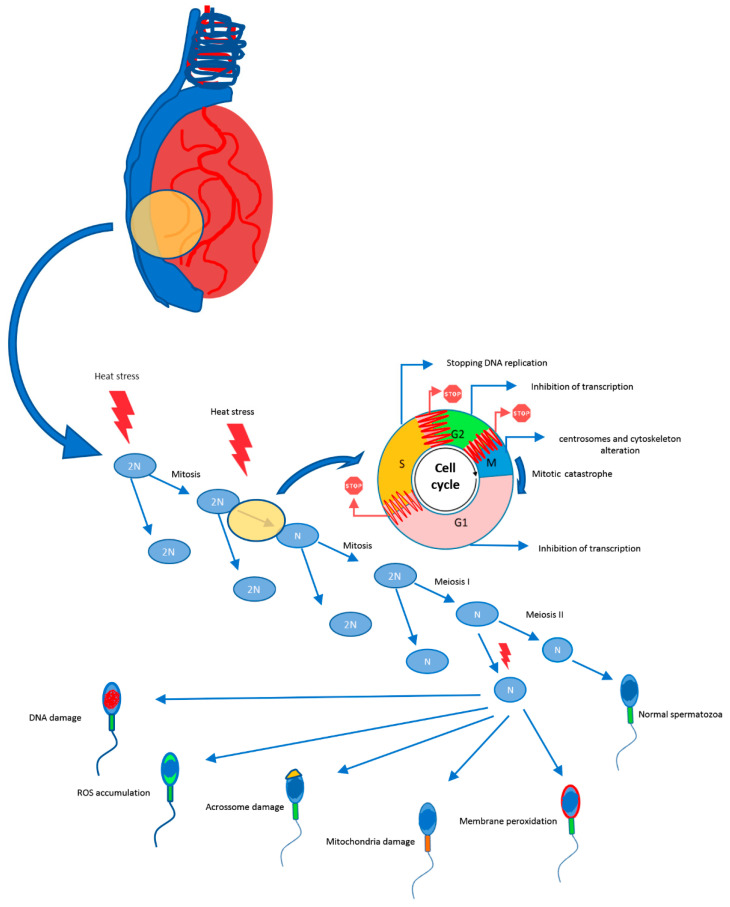
Effect of heat stress on the course of spermatogenesis.

## Data Availability

Not applicable.

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
