# Peer review of "Impact of Heat Stress on Bovine Sperm Quality and Competence"

_animals, 2022, doi:10.3390/ani12080975_

Round 1

Reviewer 1 Report

This is an interesting and well written manuscript; the information is well presented, the language is used correctly and very easy to understand.

Line 61, 94, 226: (°C) Is the correct symbol for Celsius in place of: (ºC).

It is recommend to add information about impact of heat stress on seminal plasma lipid peroxidation and sperm epigenetics.

Reviewer 2 Report

In this review, Capela and colleagues analyzed the impact of heat stress, especially due to climate changes, on bovine sperm quality. They summarized the most recent and relevant literature concerning the effects of heat stress on spermatogenesis, mainly caused by augmented oxidative stress, which negatively affects apoptosis, DNA integrity and capacitation.

The focus of the review is interesting and actual. I suggest these modifications to improve the quality of the MS prior its publication in Animals:

  • The title is a bit misleading, so I suggest to add “sperm quality” or something;
  • The developing “adaptation mechanisms” cited in line 35 are interesting, so some other details should be given;
  • In line 117, the full name and the role of RIPK1 should be added;
  • In sections 5.1, 5.2, 5.3 and 6, I suggest to add also the most used techniques to evaluate the parameter of the subheading.

Reviewer 3 Report

Animals 1641583

Impact of heat stress on bovine sperm fertilization ability

In the present Review authors made a deep revision on the impact of heat stress in fertilizing ability of bull sperm.

The topic is interesting and very actual. The revision is well-conducted and systematised. The several mechanisms (oxidative stress, DNA damage, among others) that impair male fertility are deeply analysed –especially from a biochemical point of view- and the work could be useful as starting point for other research works.

In order to become even better, several corrections should be considered.

General comments:

-An extensive English edition should be realized in order to mitigate typos in the text.

-The influence of heat stress in testicular vascularization and how the testicular vascular parameters affect fertility is not so well documented in the present review. Please see the extensive work of John P. Kastelic, a reference for this topic. For instance, in small ruminants there are adaptations in the male scrotum anatomy in order to minimise the impact of heat stress. Is there the same mechanism in bulls?

-In bulls, the spermatic cord length could have influence as well, which could be seen in Zebus. This would carry out differences in the vascular indices at the testicular artery as well.

Specific comments:

-L 38: please add a reference to the end of this sentence.

-L 43: subfertile.

-L 46: emphasizes.

-L 76: perspiration actually starts at 4ºC? Please confirm.

-L 90: how is THI, thermal humidity index, calculated?

-L 103: under heat stress there is immediate lowering of basal metabolism or there is a transitory increase of metabolism (which leads to tissue hypoxia) and then decreasing? Please verify.

Besides, latter in oxidative stress, authors confirm at L 171 that the augmentation of testicular temperature leads to increase of testicular metabolism. As the testicle has limited capacity to respond to this demand, testicular hypoxia installs soon.

These 2 affirmations should be reviewed and corrected, if necessary.

-L 187: cytoplasm.

-L 239: how does heat stress affects sperm motility? Please add some information on this topic. Again, the works of J Kastelic are excellent to address this topic.

-Conclusions: nevertheless the remarkable level of the present review, the Conclusion section was the least successful part, as there is no suggestion for future research, or perspective for heat stress impact mitigation in animals’ fertility.  
